# Does gestational diabetes increase the risk of maternal kidney disease? A Swedish national cohort study

Peter M. Barrett[ID][1,2]*, Fergus P. McCarthy[2], Marie Evans[3], Marius Kublickas[4], Ivan J. Perry[1], Peter Stenvinkel[3], Karolina Kublickiene[3‡], Ali S. Khashan[1,2‡]

1 School of Public Health, University College Cork, Cork, Ireland, 2 Irish Centre for Fetal and Neonatal Translational Research, Cork University Maternity Hospital, University College Cork, Cork, Ireland, 3 Division of Renal Medicine, Department of Clinical Intervention, Science and Technology (CLINTEC), Karolinska Institutet, Karolinska University Hospital, Stockholm, Sweden, 4 Department of Obstetrics & Gynaecology, Karolinska University Hospital, Stockholm, Sweden

‡ KK and ASK authors contributed equally and joint senior authors on this work.
* peter.barrett@ucc.ie

**Data Availability Statement:** Data are from the Swedish Medical Birth Register, National Patient Register and Swedish Renal Register. Data cannot be put into a public data repository due to Swedish

## Abstract

### Background

Gestational diabetes (GDM) is associated with increased risk of type 2 diabetes (T2DM) and cardiovascular disease. It is uncertain whether GDM is independently associated with the risk of chronic kidney disease. The aim was to examine the association between GDM and maternal CKD and end-stage kidney disease (ESKD) and to determine whether this depends on progression to overt T2DM.

### Methods

A population-based cohort study was designed using Swedish national registry data. Previous GDM diagnosis was the main exposure, and this was stratified according to whether women developed T2DM after pregnancy. Using Cox regression models, we estimated the risk of CKD (stages 3–5), ESKD and different CKD subtypes (tubulointerstitial, glomerular, hypertensive, diabetic, other).

### Findings

There were 1,121,633 women included, of whom 15,595 (1·4%) were diagnosed with GDM. Overall, GDM-diagnosed women were at increased risk of CKD (aHR 1·81, 95% CI 1·54–2·14) and ESKD (aHR 4·52, 95% CI 2·75–7·44). Associations were strongest for diabetic CKD (aHR 8·81, 95% CI 6·36–12·19) and hypertensive CKD (aHR 2·46, 95% CI 1·06–5·69). These associations were largely explained by post-pregnancy T2DM. Among women who had GDM + subsequent T2DM, strong associations were observed (CKD, aHR 21·70, 95% CI 17·17–27·42; ESKD, aHR 112·37, 95% CI 61·22–206·38). But among those with GDM only, associations were non-significant (CKD, aHR 1·11, 95% CI 0·89–1·38; ESKD, aHR 1·58, 95% CI 0·70–3·60 respectively).

confidentiality regulations for registry data. Details on the application procedures for data usage is available on the homepages of the respective registries: the Medical Birth Register (https://www.socialstyrelsen.se/en/statistics-and-data/registers/alla-register/the-swedish-medical-birth-register/); the National Patient Register (https://www.socialstyrelsen.se/en/statistics-and-data/registers/alla-register/the-national-patient-register/); and the Swedish Renal Register (https://www.medscinet.net/snr/).

**Funding:** This work was performed within the Irish Clinical Academic Training (ICAT) Programme (www.icatprogramme.org), supported by the Wellcome Trust and the Health Research Board (Grant Number 203930/B/16/Z), the Health Service Executive National Doctors Training and Planning, and the Health and Social Care, Research and Development Division, Northern Ireland. PMB is employed as an ICAT Fellow. KK is supported by the Strategic Research Programme in Diabetes at Karolinska Institutet (Swedish Research Council grant number 2009-1068 and Swedish Research Council grant number 2018-00932), Stockholm County Council (ALF), the Swedish Kidney Foundation (Njurfonden). ME is supported by the Center of Innovate Medicine (CIMED) at Karolinska Institutet and ALF Medicine. The funders had no role in study design, data collection and analysis, decision to publish, or preparation of the manuscript.

**Competing interests:** I have read the journal's policy and the authors of this manuscript have the following competing interests: ME has participated in advisory board meetings (Astellas, Astra Zeneca, Vifor Pharma) and has received payment for lectures (Astellas, Vifor Pharma). This does not alter our adherence to PLOS ONE policies on sharing data and materials.

## Conclusion

Women who experience GDM and subsequent T2DM are at increased risk of developing CKD and ESKD. However, GDM-diagnosed women who never develop overt T2DM have similar risk of future CKD/ESKD to those with uncomplicated pregnancies.

## Background

The incidence of gestational diabetes (GDM) is increasing and in 2019, it was estimated that 13% of all pregnancies worldwide may be affected by GDM [1]. The reasons for this are multi-factorial, but they include rising maternal age, a higher prevalence of obesity among pregnant women, and lowering of diagnostic thresholds for GDM [2]. Women who experience GDM are at higher risk of later cardiovascular morbidity and mortality [3], but less is known about the long-term renal sequelae following GDM. Chronic kidney disease (CKD) is a highly prevalent and preventable cause of ill-health among women, and its incidence is higher among those who experience other pregnancy-related complications [4]. End-stage kidney disease (ESKD), although relatively rare, causes a disproportionate burden of morbidity and premature mortality [5]. However, few studies have examined whether GDM independently increases the risk of maternal CKD or ESKD [6–9].

Women who experience GDM have a 10-fold increased risk of developing type 2 diabetes (T2DM) [10], and T2DM is a risk factor for CKD [2]. Yet, about half of GDM-diagnosed women will never develop T2DM. GDM-diagnosed women are more likely to have persistent markers of endothelial dysfunction in the years following pregnancy [11–13]. As a result, they may be predisposed to a range of cardiovascular and renal diseases compared to women who remained normoglycaemic in pregnancy. But to date, the evidence linking prior GDM diagnosis to long-term risk of renal impairment has been conflicting [4, 7, 14–18].

The aim of this study was to measure the association between GDM and subsequent risk of maternal CKD and ESKD. We sought to determine whether this risk persists across a range of CKD subtypes independent of medical and obstetric comorbidities. We also sought to identify whether this risk differed among GDM-diagnosed women according to whether they were subsequently diagnosed with T2DM, or not.

## Methods

### Study design

We undertook a nationwide, population-based cohort study of mothers who gave birth in Sweden between 01/01/87 and 31/12/12. Data were obtained from the Swedish Medical Birth Register (MBR, established 1973) and women were included if they had their first birth on or after 01/01/87, when information on GDM diagnosis was available. The MBR was validated in 2002, and the quality of variables was deemed to be high [19].

The information from the MBR was linked to data from the Swedish National Patient Register (NPR) and Swedish Renal Register (SRR) up to 31/12/13 to identify those who developed CKD or ESKD during follow-up. The NPR contained information on inpatient admissions from 1964 onwards and outpatient reviews from 2001 onwards. The SRR contained information on ESKD diagnoses from 1991 onwards and on CKD (stage 3–5) from 2007 onwards. The SRR mainly recorded outpatient visits for patients with CKD once they reached an eGFR <30mL/min/1.73m$^2$ (i.e. stage 4–5 CKD). However, nephrology units are also encouraged to

include patients who are earlier in the course of their disease (eGFR 30-60mL/min/1.73m$^2$, i.e. stage 3 CKD), but this is not mandatory. The registers were linked using unique anonymised serial numbers (lpnr) which were derived from each participant's personal identification number before the research team received the data. Information from the Swedish Death Register and Migration Register were also used to censor women who died or emigrated during follow-up.

We sought to include a healthy population at baseline to reduce the possibility of confounding by comorbidities. Women were excluded at baseline if they had pre-pregnancy medical conditions which may increase the risk of CKD/ESKD. We used the MBR and NPR to identify, and exclude, women who had the following comorbidities on or before the date of their first delivery: previous CKD/ESKD, cardiovascular disease (CVD), chronic hypertension, systemic lupus erythematosus, systemic sclerosis, coagulopathies, haemoglobinopathies, or vasculitides. Women who had a diagnosis of diabetes at baseline (type 1 or type 2), defined as a diagnosis in the MBR or NPR on or before the date of their first delivery, were also excluded since those women could not, by default, develop GDM within the dataset (S1 Fig).

We excluded women who had multiple pregnancies, births with implausible dates of delivery, and births with implausible birth weights for gestational age. We used three iterations of ICD coding to identify women who had pre-pregnancy disease; ICD-8 coding (1973–1986, used for checking previous diagnoses from the NPR and MBR at baseline), ICD-9 coding (1987–1996 inclusive) and ICD-10 coding (1997 onwards). The list of ICD codes used in the study is available in S1 Table.

## Gestational diabetes

GDM was the main exposure of interest and was based on ICD-coded diagnosis in the MBR or NPR. ICD coding for GDM was only available from 1987 onwards (ICD-9 code 648W, ICD-10 code O244), hence the study was restricted to women whose first birth occurred during or after 1987.

In Sweden, there has been lack of consensus regarding screening regimes for GDM. Antenatal care is organised across 43 different Maternal Health Care Areas (MHCAs). Some MHCAs apply universal screening of GDM to all pregnant women and other regions use a selective approach based on particular risk factors (e.g. previous GDM, previous stillbirth, body mass index (BMI) $\geq$30 kg/m$^2$, macrosomic infant >4.5kg) or random blood glucose measurements [20]. Both universal and selective screening regimes stipulate the use of a 75-g oral glucose tolerance test and 2-hour value of capillary plasma glucose for diagnosis, but the diagnostic thresholds for GDM vary across MHCAs. The 2-hour plasma glucose diagnostic thresholds ranged from 9.0–11.1 mmol/L during the study period, using either capillary or venous samples. One-third of MHCAs also used fasting glucose as a diagnostic criterion for GDM. If a fasting threshold was used, then GDM was based on fasting thresholds of 6.1–7.0 mmol/L [20, 21]. We did not have information in the MBR or NPR on the specific diagnostic thresholds used for individuals.

Two separate exposure variables were used for GDM. The first variable was dichotomous (any GDM vs. none) and it was included in statistical models as a time-dependent variable. Women were considered 'exposed' from the date of their first delivery with GDM, irrespective of subsequent unaffected pregnancies. Women were considered unexposed (i) if they never developed GDM, or (ii) from the date of their first delivery (without GDM) until the date of their first GDM-affected delivery. Thus, if a woman had an unaffected pregnancy first (without GDM) and was diagnosed with GDM during a subsequent pregnancy, she would contribute both unexposed and exposed person-time during follow-up.

A second time-dependent exposure variable was created to further categorise GDM-diagnosed women according to whether they developed overt T2DM. Diagnoses of T2DM were identified from the MBR and NPR using ICD coding. The following categories were used: (i) neither GDM nor T2DM (reference group) (ii) GDM-diagnosed only, no subsequent T2DM (iii) T2DM-diagnosed only, no prior GDM (iv) GDM first + subsequent T2DM. This approach was consistent with a previous large-scale cohort study of GDM and maternal CVD [22].

Large for gestational age (LGA) was considered as a proxy marker of GDM severity if it occurred in the same pregnancy as GDM [23]. LGA was defined in the MBR as a birth weight of 2 standard deviations (SD) above the sex-specific and gestational age distributions, according to Swedish weight-based growth standards [24].

## Outcome variables

Maternal CKD and ESKD were the main outcomes defined by a verified diagnosis in the NPR or SRR. ESKD was defined as stage 5 CKD, requiring dialysis or renal transplant. The earliest date at which a woman appeared in either the NPR or the SRR was assumed to be her date of diagnosis for CKD/ESKD. Women were excluded if they were diagnosed with CKD/ESKD within three months of their last pregnancy to avoid potential misclassification with acute kidney injury. Women who had any form of CKD/ESKD due to an identifiable congenital or genetic cause were also excluded to reduce the possibility of confounding (S1 Table). The following subtypes/aetiologies of CKD were used: tubulointerstitial, glomerular/proteinuric (i.e. nephrotic syndrome, nephritic syndrome, chronic glomerulonephritis), hypertensive, diabetic, other/unspecified CKD. The process for selecting these categories has been described in detail elsewhere [25].

## Covariates

We adjusted for the following covariates: maternal age, country of origin (Sweden vs elsewhere), maternal education (highest level achieved), parity, antenatal BMI in first pregnancy, gestational weight gain, smoking during pregnancy and preeclampsia. All analyses were stratified by year of delivery. Information on maternal education was based on the highest educational achievement recorded in the Swedish Register of Education. Smoking status was based on any reported smoking during pregnancy, either at first antenatal visit or at 30–32 weeks' gestation. BMI was measured based on weight (kg) and height (m) at first antenatal visit. Gestational weight gain was measured by subtracting each woman's weight (kg) at first antenatal visit from her weight (kg) at the time of first delivery. This was categorised as optimal, inadequate or excessive using established criteria, based on BMI category at first antenatal visit [26]. Maternal exposure to preeclampsia was included as a time-dependent covariate. Preeclampsia was defined as a diastolic blood pressure of >90 mm Hg with proteinuria (0.3 g/d or $\geq$1+ on a urine dipstick), but excluding women who developed preeclampsia superimposed on chronic hypertension since women with pre-pregnancy hypertension were excluded at baseline [25].

## Ethical considerations

Ethical approval was obtained from the Swedish Ethical Review Authority in Stockholm (Regionala Etikprövningsnämnden Stockholm; Dnr 2012/397-31/1) and the Social Research and Ethics Committee, University College Cork (2019–109).

## Statistical analysis

The association between GDM and risk of maternal CKD/ESKD was measured using the Kaplan-Meier method. The log-rank test was used to estimate differences in survival curves.

Multivariable Cox proportional hazard regression models were used to estimate age-adjusted and fully adjusted hazard ratios (aHRs) and 95% confidence intervals (CI). Women were followed up from the date of their first singleton birth until date of diagnosis of CKD/ESKD, study end date (31/12/13), or censoring due to death or emigration, whichever came first. Thus, women stopped contributing person-time once they were diagnosed with CKD or ESKD, and any subsequent pregnancies were not included in the analysis. Two-sided p-values were used, and p <0.05 denoted statistical significance.

Firstly, we estimated the overall association between GDM and maternal CKD and ESKD respectively (vs. women who never had GDM, irrespective of subsequent T2DM). Secondly, we measured associations between GDM and subtypes of CKD in separate models. Thirdly, we repeated the analyses to identify whether associations differed according to whether GDM-diagnosed women developed subsequent T2DM. We also explored the associations between GDM +/- LGA with maternal CKD and ESKD respectively. *A priori*, we planned to assess effect modification by country of birth (Sweden vs. elsewhere) and maternal BMI (obese vs. non-obese at first antenatal visit). All analyses were performed using Stata version 15 (Stata-Corp LLC).

There were missing data for maternal education, smoking, BMI at first antenatal visit, and gestational weight gain [19, 27]. We used multiple imputation by chained equations to address missing data, using linear models to impute BMI and gestational weight gain, and multinomial logistic models to impute maternal education and smoking status (M = 20).

## Results

The study cohort consisted of 1,121,633 unique women who had 2,458,580 singleton births, followed up for a total of 15,303,798 person-years. The median follow-up time was 12.2 years (interquartile range (IQR) 6.2 to 19.3 years).

There were 15,595 women (1.4%) diagnosed with GDM at least once. The incidence of GDM doubled over time, from 562 diagnoses per 100,000 births in 1987–1991, to 1,116 diagnoses per 100,000 births in 2007–2012. The demographic profile of pregnant women also changed over time from mean age 26.4 (± 4.7) years in 1987–1991, and 3.9% prevalence of obesity, to mean maternal age 30.2 (± 5.3) years in 2007–2012, and 12.1% prevalence of obesity (S2 Table). Women who were diagnosed with GDM were more likely to be older in age at first delivery, born outside of Sweden, overweight or obese at first antenatal visit, and were less likely to have a third level education than other women (Table 1).

From 1987 to 2013, 5,879 women (0.5%) developed CKD, of whom 1,343 (22.8%) had tubulo-interstitial CKD; 1,800 (30.6%) had glomerular/proteinuric CKD; 138 (2.3%) had hypertensive CKD; 137 (2.3%) had diabetic CKD; 2,461 (41.8%) had CKD due to other/unspecified causes. Overall, 228 women (0.02%) developed ESKD during follow-up. There were 12,232 deaths during follow-up (1.1% of all participants). GDM diagnosis was not significantly associated with mortality during follow-up (p = 0.804). However, the mortality rate was significantly higher among women who were diagnosed with CKD during follow-up (n = 349, 5.9%) compared with women were not diagnosed with CKD (n = 11,883, 1.1%) (p<0.001).

### Any history of gestational diabetes

The risk of CKD appeared to be increased among women who ever had a history of GDM compared with women who did not (log-rank p <0.001). After adjusting for potential confounders, women who were ever diagnosed with GDM had higher risk of developing CKD (vs. no GDM, aHR 1.81, 95% CI 1.54–2.14) (Table 2). The median time to CKD diagnosis was also

**Table 1. Maternal characteristics and pregnancy outcomes among women whose first birth occurred between 1987 and 2012 in Sweden, stratified by exposure to GDM and/or type 2 diabetes (n = 1,121,633).**

|  | No GDM or T2DM, n (%) | GDM only, n (%) | T2DM only, n (%) | GDM & T2DM, n (%) |
|---|---|---|---|---|
|  | N = 1,104,488 (98·5) | N = 14,751 (1·3) | N = 1,550 (0·1) | N = 844 (0·1) |
| **Age at first delivery (years)** |  |  |  |  |
| *Mean ± sd* | *27·5 ± 5·1* | *28·6 ± 5·6* | *26·3 ± 4·9* | *26·5 ± 5·0* |
| **Native country** |  |  |  |  |
| Sweden | 916,776 (83·0) | 10,188 (69·1) | 1,288 (83·1) | 618 (73·2) |
| Elsewhere | 187,712 (17·0) | 4,563 (30·9) | 262 (16·9) | 226 (26·8) |
| **Education level** |  |  |  |  |
| Less than Upper Secondary | 104,159 (9·4) | 2,312 (15·7) | 206 (13·3) | 123 (14·6) |
| Upper Secondary | 485,675 (44·0) | 6,735 (45·7) | 749 (48·3) | 421 (49·9) |
| Third level | 514,654 (46·6) | 5,704 (38·7) | 595 (38·4) | 300 (35·6) |
| **BMI in early pregnancy (kg/m$^2$)** |  |  |  |  |
| Underweight: <18·5 | 65,327 (5·9) | 583 (4·0) | 90 (5·8) | 39 (4·6) |
| Normal: 18·5–24·9 | 715,438 (64·8) | 6,548 (44·4) | 808 (52·1) | 393 (46·6) |
| Overweight: 25–29·9 | 245,523 (22·2) | 4,394 (29·8) | 413 (26·7) | 246 (29·2) |
| Obese: ≥30 | 78,200 (7·1) | 3,226 (21·9) | 239 (15·4) | 166 (19·7) |
| **Gestational weight gain*** |  |  |  |  |
| Optimal | 216,008 (19·6) | 3,785 (25·7) | 293 (19·0) | 205 (24·4) |
| Inadequate | 9,658 (0·9) | 120 (0·8) | 10 (0·6) | 10 (1·2) |
| Excessive | 877,271 (79·5) | 10,817 (73·5) | 1,242 (80·4) | 627 (74·5) |
| **Maternal smoking** |  |  |  |  |
| Yes | 159,127 (14·4) | 2,171 (14·7) | 297 (19·2) | 133 (15·8) |
| No | 945,361 (85·6) | 12,580 (85·3) | 1,253 (80·8) | 711 (84·2) |
| **Preeclampsia (ever)** |  |  |  |  |
| Yes | 52,682 (4·8) | 1,610 (10·9) | 230 (14·8) | 118 (14·0) |
| No | 1,051,806 (95·2) | 13,141 (89·1) | 1,320 (85·2) | 726 (86·0) |
| **Large for gestational age (ever)** |  |  |  |  |
| Yes | 55,766 (5·1) | 2,860 (19·4) | 578 (37·3) | 275 (32·6) |
| No | 1,048,247 (94·8) | 11,888 (84·6) | 971 (62·7) | 569 (67·4) |
| **Small for gestational age (SGA) (ever)** |  |  |  |  |
| Yes | 48,988 (4·4) | 603 (4·1) | 62 (4·0) | 49 (5·8) |
| No | 1,055,025 (95·6) | 14,145 (95·9) | 1,487 (96·0) | 795 (94·2) |
| **Stillbirth (ever)** |  |  |  |  |
| Yes | 3,524 (0·3) | 135 (0·9) | 13 (0·8) | 11 (1·3) |
| No | 1,100,964 (99·7) | 14,616 (99·1) | 1,537 (99·2) | 833 (98·7) |

BMI, body mass index; GDM, gestational diabetes; T2DM, type 2 diabetes (diagnosed after the first delivery). Women who had any diagnosis of type 1 or type 2 diabetes mellitus before or during their first pregnancy were excluded. These results are based on multiple imputation due to missing data on maternal smoking, BMI in early pregnancy, gestational weight gain, and education level.

*Categories as defined by Cedergren et al. (26)

shorter in women who were exposed to GDM (median 5.9 years, IQR 2.6–12.0) compared to women who were never diagnosed with GDM (median 6.7 years, IQR 2.8–11.4) (S3 Table).

The risk of CKD differed considerably by subtype. The association was particularly strong for diabetic CKD (aHR 8.81, 95% CI 6.36–12.19), but was also observed for hypertensive CKD (aHR 2.46, 95% CI 1.06–5.69) and glomerular CKD (aHR 1.86, 95% CI 1.37–2.51). There was no significant association between GDM and risk of future tubulo-interstitial CKD or other/non-specific forms of CKD. GDM was associated with increased risk of ESKD (vs. no GDM,

**Table 2. Hazard ratios for maternal chronic kidney disease and end-stage kidney disease by history of GDM among women whose first birth occurred between 1987 and 2012 in Sweden (n = 1,121,633).**

| | | Chronic kidney disease (N = 5,879) | |
|---|---|---|---|
| | **n** | **Age-adjusted** | **Fully adjusted** |
| **Ever had GDM** | | **HR (95% CI)** | **HR (95% CI)** |
| None | 5,725 | 1·0 | 1·0 |
| GDM | 154 | 2·39 (2·03–2·80) | 1·81 (1·54–2·14) |
| | | *Tubulo-interstitial CKD* | |
| None | 1,325 | 1·0 | 1·0 |
| GDM | 18 | 1·28 (0·80–2·04) | 0·98 (0·62–1·57) |
| | | *Glomerular CKD* | |
| None | 1,755 | 1·0 | 1·0 |
| GDM | 45 | 2·47 (1·84–3·32) | 1·86 (1·37–2·51) |
| | | *Hypertensive CKD* | |
| None | 132 | 1·0 | 1·0 |
| GDM | 6 | 3·94 (1·76–8·96) | 2·46 (1·06–5·69) |
| | | *Diabetic CKD* | |
| None | 88 | 1·0 | 1·0 |
| GDM | 49 | 15·90 (11·71–21·59) | 8·81 (6·36–12·19) |
| | | *Other/non-specific CKD* | |
| None | 2,425 | 1·0 | 1·0 |
| GDM | 36 | 1·30 (0·94–1·81) | 1·06 (0·76–1·48) |
| | | **End-stage kidney disease (N = 228)** | |
| None | 210 | 1·0 | 1·0 |
| GDM | 18 | 6·95 (4·29–11·26) | 4·52 (2·75–7·44) |

CKD, chronic kidney disease; GDM, gestational diabetes.

Hazard ratios represent separate Cox regression models for associations between GDM and maternal chronic kidney disease, subtypes of chronic kidney disease, or end-stage kidney disease respectively. In all models, GDM was a time-dependent variable, where maternal exposure status was based on the date of first affected delivery.

Fully adjusted models were adjusted for maternal age, country of origin, maternal education, parity, antenatal BMI, smoking, gestational weight gain and maternal exposure to preeclampsia (time-dependent covariate), stratified by year of delivery. Women with pre-pregnancy history of renal disease, cardiovascular disease, diabetes, hypertension, systemic lupus erythematosus, coagulopathies, haemoglobinopathies and vasculitis were excluded at baseline.

aHR 4.52, 95% CI 2.75–7.44), but there were too few ESKD outcomes to allow a separate analysis of ESKD subtypes.

Women who experienced GDM and LGA concurrently had a higher risk of CKD (aHR 3.03, 95% CI 2.28–4.03) compared with those who experienced GDM alone (without LGA) (aHR 1.58, 95% CI 1.31–1.93). Similarly, the risk of ESKD was stronger in women who experienced GDM and LGA concurrently (aHR 8.37, 95% CI 3.64–19.23) than in women who experienced GDM alone (aHR 3.78, 95% CI 2.08–6.87) (S4 Table).

## Gestational diabetes and subsequent type 2 diabetes

When the effect of subsequent T2DM was considered, the associations between GDM (only) and CKD or ESKD were attenuated to non-significance (CKD, aHR 1.11, 95% CI 0.89–1.38; ESKD, aHR 1.58, 95% CI 0.70–3.60 respectively) (Table 3). Women who had a history of T2DM alone had increased risk of CKD (aHR 20.70, 95% CI 18.72–22.88), ESKD (aHR 59.56, 95% CI 42.90–82.70), and each of the renal subtypes. Women who had been first diagnosed

**Table 3. Hazard ratios for maternal chronic kidney disease and end-stage kidney disease by history of GDM and/or type 2 diabetes, among women whose first birth occurred between 1987 and 2012 in Sweden (n = 1,121,633).**

| | | Chronic kidney disease (N = 5,879) | |
|---|---|---|---|
| | | Age-adjusted | Fully adjusted |
| **Ever had GDM or T2DM** | | **HR (95% CI)** | **HR (95% CI)** |
| None | 5,559 | 1·0 | 1·0 |
| GDM only | 81 | 1·40 (1·13–1·75) | 1·11 (0·89–1·38) |
| T2DM only | 166 | 24·36 (22·07–26·89) | 20·70 (18·72–22·88) |
| GDM + T2DM | 73 | 33·57 (26·63–42·32) | 21·70 (17·17–27·42) |
| | | *Tubulo-interstitial CKD* | |
| None | 1,315 | 1·0 | 1·0 |
| GDM only | 17 | 1·27 (0·79–2·05) | 1·00 (0·61–1·61) |
| T2DM only | 10 | 5·18 (3·37–7·99) | 4·55 (2·95–7·02) |
| GDM + T2DM | <5 | ne | ne |
| | | *Glomerular CKD* | |
| None | 1,734 | 1·0 | 1·0 |
| GDM only | 36 | 2·10 (1·51–2·92) | 1·60 (1·15–2·24) |
| T2DM only | 21 | 8·20 (6·11–10·99) | 6·67 (4·97–8·97) |
| GDM + T2DM | 9 | 14·22 (7·38–27·41) | 8·80 (4·55–17·01) |
| | | *Hypertensive CKD* | |
| None | 121 | 1·0 | 1·0 |
| GDM only | <5 | ne | ne |
| T2DM only | 11 | 38·55 (22·41–66·32) | 27·92 (16·02–48·68) |
| GDM + T2DM | 5 | 102·81 (41·72–253·39) | 56·81 (22·36–144·35) |
| | | *Other/non-specific CKD* | |
| None | 2,389 | 1·0 | 1·0 |
| GDM only | 27 | 1·04 (0·71–1·51) | 0·86 (0·58–1·26) |
| T2DM only | 36 | 7·32 (5·61–9·54) | 6·42 (4·90–8·4) |
| GDM + T2DM | 9 | 9·17 (4·77–17·67) | 6·36 (3·30–12·27) |
| | | **End-stage kidney disease (N = 228)** | |
| None | 191 | 1·0 | 1·0 |
| GDM only | 6 | 3·05 (1·35–6·88) | 1·58 (0·70–3·60) |
| T2DM only | 19 | 79·43 (57·84–109·09) | 59.56 (42·90–82.70) |
| GDM + T2DM | 12 | 163·37 (90·53–294·82) | 112·37 (61·22–206·38) |

CKD, chronic kidney disease; GDM, gestational diabetes.

Hazard ratios represent separate Cox regression models for associations between GDM and/or type 2 diabetes (diagnosed after the first delivery) and maternal CKD, subtypes of CKD, or end-stage kidney disease respectively. Diabetic CKD was excluded from this table because nobody in the reference group (i.e. never diagnosed with GDM nor T2DM) could develop the outcome which was dependent on progression to overt T2DM.

Fully adjusted models were adjusted for maternal age, country of origin, maternal education, parity, antenatal BMI, smoking, gestational weight gain and maternal exposure to preeclampsia (time-dependent covariate), stratified by year of delivery. Women with pre-pregnancy history of renal disease, cardiovascular disease, diabetes, hypertension, systemic lupus erythematosus, coagulopathies, haemoglobinopathies and vasculitis were excluded at baseline.

with GDM and subsequently developed overt T2DM during follow-up were at highest risk of future CKD (aHR 21.70, 95% CI 17.17–27.42) or ESKD (aHR 112.37, 95% CI 61.22–206.38).

Effect modification was observed by country of birth for associations with CKD. However, this was largely driven by differential associations between T2DM and CKD, rather than by GDM. No significant effect modification was observed by maternal BMI (S5 Table).

## Discussion

We aimed to determine whether GDM was independently associated with the long-term risk of maternal CKD and ESKD. Overall, women who were ever diagnosed with GDM appeared to be at higher risk of CKD/ESKD during follow-up, and this risk differed by CKD subtype. However, this was largely explained by the predisposition of GDM-diagnosed women to future T2DM. When the effects of GDM and later T2DM were separated, associations between GDM and CKD/ESKD, in the absence of subsequent T2DM, were largely attenuated and became non-significant. By contrast, strong associations persisted between T2DM and CKD/ESKD.

Previous research has reported that women with a history of GDM are more likely to have early signs of renal impairment such as elevated glomerular filtration rate [16] or microalbuminuria [7, 17] during the post-reproductive years. However, there has been uncertainty in the published literature over whether GDM independently increases the risk of clinically significant CKD (stage 3 or greater) [4]. Existing studies have been limited by incomplete adjustment for confounders like maternal obesity or pre-existing comorbidities [8, 17], use of non-specific outcome data such as renal-related hospitalisations [18], or inadequate consideration for the effects of subsequent T2DM [8, 17, 18]. Our findings are consistent with prospective studies from North America which reported that women who experience GDM, but who never develop overt T2DM, have an equivalent risk of clinically significant CKD/ESKD to those who remain normoglycaemic in pregnancy [7, 9]. The only exception to this was the risk of glomerular CKD, where a modest association persisted among women who were exposed to GDM alone. To our knowledge, our research is the first to report associations for both CKD and ESKD separately for the same cohort of women, as well as being the first to provide detailed information on CKD subtypes.

GDM has been established as an independent risk factor for subclinical inflammation [28] and endothelial dysfunction [11], but the long-term implications of this are still emerging. GDM-diagnosed women are at higher risk of metabolic syndrome in later life, even if they remain glucose-tolerant in the years following pregnancy, and this suggests an underlying predisposition to chronic disease [29]. Cardiovascular research has indicated that GDM-diagnosed women may remain at risk of future CVD irrespective of T2DM [9, 30], although the evidence for this remains inconclusive [22]. By contrast, our study suggests that GDM-diagnosed women are only at heightened risk of CKD/ESKD if they develop T2DM in the years following pregnancy. Our findings support the hypothesis that GDM may differentially impact on the long-term risk of microvascular and macrovascular disease outcomes [9].

Overall, the proportion of women diagnosed with GDM in this study was lower than expected (1.4%), and there are several possible reasons for this. The true level of GDM depends on the screening method employed (universal vs. selective), diagnostic threshold used, background characteristics of pregnant women in the population, and uptake of screening [20]. During the study period, most MHCAs in Sweden used a selective, high-risk screening approach for GDM. Diagnostic criteria for GDM were relatively strict, particularly during the earlier years of the study, and it is likely that many cases of GDM went undiagnosed. Furthermore, the prevalence of obesity was low in this study by international standards, suggesting that women of childbearing age in Sweden may have been at lower risk of GDM [31]. Nonetheless, we observed an increase in the incidence rate of GDM over the lifetime of this study. This is likely to have been driven by increases in maternal age at delivery, rising prevalence of obesity, sedentary lifestyle among some pregnant women, and changes to the diagnostic criteria [20, 31, 32].

We also examined whether concurrent LGA impacted on associations between GDM and CKD/ESKD. Mothers of LGA offspring tend to have less favourable anthropometric, lipid and

glucose levels throughout their life course, suggesting poorer metabolic health when compared with women whose offspring are born appropriate for gestational age [33]. Co-occurring GDM and LGA was associated with increases in the overall risk of future CKD/ESKD, possibly due to increased risk of progression to T2DM. The proportion of births affected by LGA is increasing, and this may be related to increases in maternal BMI, rising incidence of GDM, and decreases in maternal smoking [34]. Although all GDM-diagnosed women warrant post-partum surveillance for T2DM, those who have concurrent LGA deliveries may be at particularly high risk of chronic disease and may benefit most from earlier preventive interventions.

It has been suggested that GDM-diagnosed black women may be at higher risk of CKD than GDM-diagnosed white women [4]. We were only able to explore ethnicity effects in this study according to maternal country of birth (Sweden vs. elsewhere). Sweden had a predominantly white Caucasian population during the study period, and it is likely that those born outside of Sweden were of wider ethnic diversity. However, our analysis suggests that any effect modification by ethnic origin may be driven by differential associations between T2DM and CKD, and not by GDM itself. Black women have an increased risk of T2DM compared with white women [35], and this may increase their risk of future CKD irrespective of previous GDM [36, 37].

Women who experienced T2DM with or without previous GDM were at increased risk of CKD and ESKD, including multiple subtypes of renal pathology unrelated to diabetes. The mechanisms underlying the associations with non-diabetic CKD are uncertain. There may be shared inflammatory or metabolic regulatory pathways which lead to CKD progression in hyperglycaemic women. For example, decreased expression of renal nuclear factor erythroid 2-related factor 2 (NRF 2) may increase the risk of a range of kidney diseases in later life [38].

## Strengths and limitations

The national Swedish registers have a high level of completeness, and contain data on >96% of pregnant women [19]. We were able to adjust for a wide range of covariates, and we reduced confounding by excluding women with relevant comorbidities, including pre-pregnancy diabetes and renal disease. We classified CKD according to specific subtypes to get a more detailed overview of both diabetic and non-diabetic forms of CKD, and we were able to separate the effects of GDM from T2DM depending on the timing of each diagnosis.

Our information on GDM was based on ICD-coded diagnosis, and we were unable to identify which screening or diagnostic criteria had been applied for different individuals. Most MHCAs employed a selective, high-risk approach to screening and this may have introduced differential misclassification since obese women, those with LGA deliveries, and women with a prior history of GDM were more likely to have been diagnosed with GDM compared with those who appeared otherwise healthy. Many cases of GDM may have been undiagnosed and this may have diluted the magnitude of true effect sizes [21]. However, we considered the relevant screening criteria in our statistical models (e.g. obesity, LGA) and thus, the risk of CKD/ESKD is unlikely to differ substantially from that observed here. In 2015, the Swedish National Board of Health and Welfare recommended a move to standardised WHO diagnostic criteria for GDM using venous sampling [39], but this occurred after the study period.

Most cases of CKD/ESKD were identified using ICD-coded diagnoses in the NPR, with fewer cases identified from the SRR. Although it is likely that ESKD data were virtually complete, women with CKD may have been under-diagnosed or under-ascertained. Although the NPR had achieved national coverage for all hospital admissions in Sweden by 1987, outpatient review data were only collected from 2001 onwards. The SRR began to collect ESKD data from 1991 onwards, and only collected CKD data from 2007. Some mothers may have been too

young to have developed symptomatic CKD, particularly for hypertensive or diabetic subtypes which tend to develop over decades and affect women in later life. The burden of subclinical pathology among these women is uncertain. However, those women who were identified as CKD/ESKD cases were likely to have valid diagnoses given that most diagnoses in the NPR have high positive predictive values [40].

Although we controlled for a wide variety of covariates, we had no information on specific treatments administered for GDM, lifestyle factors (e.g. diet, physical activity), nor biomarker data such as glucose tolerance status, glomerular filtration rate, or dyslipidaemia at follow-up. Furthermore, while we excluded women who had inherited forms of CKD at baseline, we had no information on family history of T2DM, thus we cannot exclude the possibility of some residual genetic confounding.

The proportion of women who progressed from GDM to T2DM was low. This may have been because our cohort mainly consisted of Caucasian women who were healthy at baseline. The period of follow-up after pregnancy was relatively short, and T2DM may have been under-ascertained since we only had access to hospital-level diagnoses, and not primary care data. Moreover, given that we excluded women with relevant medical comorbidities at baseline, we were unable to assess the potential additive effect of GDM in women who may be otherwise predisposed to CKD.

These findings suggest that women who experience GDM and subsequent T2DM may be at significantly higher risk of CKD and ESKD, but women who experience GDM alone (without later T2DM) have equivalent risk of future kidney disease to those whose pregnancies were normoglycaemic. Postpartum screening for T2DM, and lifestyle or pharmacological interventions aimed at preventing onset of T2DM, are likely to reduce the burden of kidney disease among women who have been diagnosed with GDM. Given that GDM disproportionately affects marginalised groups, including women from less educated or ethnic minority backgrounds [1, 2], such interventions may help to reduce existing health inequalities.

It has been suggested that obstetric information may add incremental value to clinical risk prediction tools for CVD or CKD [41]. International cardiovascular guidelines now suggest that adverse pregnancy outcomes may be considered as sex-specific 'risk enhancing factors' which can be used to inform primary and secondary preventive interventions for patients in lower CVD risk categories [42]. However, to date there is a dearth of research on the use of obstetric risk factors in renal risk prediction models. Further population-based studies with long periods of follow up (>10 years) may be needed to determine whether obstetric information, including GDM, can be used to enhance cardiometabolic risk prediction algorithms for women going forward [41].

## Conclusion

Women who experience GDM may be at increased risk of CKD or ESKD in later life, but this is largely explained by their predisposition to T2DM in the intervening years. GDM-diagnosed women who do not develop subsequent T2DM appear to have an equivalent risk of future CKD/ESKD to those who remain normoglycaemic in pregnancy. Postpartum screening for T2DM, and lifestyle or pharmacological interventions aimed at preventing onset of T2DM, are likely to reduce the burden of kidney disease among women affected by GDM.

## Supporting information

**S1 Table. ICD codes used for disease definitions.**
(DOCX)

**S2 Table. Demographic changes over time among women whose first delivery occurred in Sweden between 1987 and 2012.**
(DOCX)

**S3 Table. Time to diagnosis of chronic kidney disease (CKD) subtypes among women whose first live birth occurred between 1987 and 2012 in Sweden, stratified by exposure to GDM.**
(DOCX)

**S4 Table. Hazard ratios for maternal kidney disease by history of gestational diabetes and delivery of a large for gestational age infant, among women whose first birth occurred between 1987 and 2012 in Sweden.**
(DOCX)

**S5 Table. Effect modification by ethnicity and antenatal obesity status of the association between gestational diabetes and/or type 2 diabetes and maternal renal disease in women whose first birth occurred between 1987 and 2012 in Sweden.**
(DOCX)

**S1 Fig. Flow chart illustrating construction of study cohort.**
(DOCX)

## Author Contributions

**Conceptualization:** Peter M. Barrett, Fergus P. McCarthy, Marie Evans, Marius Kublickas, Ali S. Khashan.

**Data curation:** Marius Kublickas, Karolina Kublickiene.

**Formal analysis:** Peter M. Barrett.

**Funding acquisition:** Peter M. Barrett, Karolina Kublickiene.

**Investigation:** Peter M. Barrett, Ali S. Khashan.

**Methodology:** Peter M. Barrett, Fergus P. McCarthy, Marie Evans, Marius Kublickas, Peter Stenvinkel, Karolina Kublickiene, Ali S. Khashan.

**Project administration:** Marius Kublickas, Karolina Kublickiene, Ali S. Khashan.

**Software:** Marius Kublickas.

**Supervision:** Fergus P. McCarthy, Ivan J. Perry, Karolina Kublickiene, Ali S. Khashan.

**Writing – original draft:** Peter M. Barrett.

**Writing – review & editing:** Peter M. Barrett, Fergus P. McCarthy, Marie Evans, Marius Kublickas, Ivan J. Perry, Peter Stenvinkel, Karolina Kublickiene, Ali S. Khashan.

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
