## [Decision Letter · Decision Letter 0]

16 Nov 2021

PONE-D-21-31275Does gestational diabetes increase the risk of maternal kidney disease? A Swedish national cohort studyPLOS ONE

Dear Dr. Barrett,

Thank you for submitting your manuscript to PLOS ONE. After careful consideration, we feel that it has merit but does not fully meet PLOS ONE’s publication criteria as it currently stands. Therefore, we invite you to submit a revised version of the manuscript that addresses the points raised during the review process.

We look forward to receiving your revised manuscript.

Kind regards,

Forough Mortazavi

Academic Editor

PLOS ONE

Journal Requirements:

"I have read the journal’s policy and the authors of this manuscript have the following competing interests: ME has participated in advisory board meetings (Astellas, Astra Zeneca, Vifor Pharma) and has received payment for lectures (Astellas, Vifor Pharma)."

Additional Editor Comments:

Dear Authors,

Thank you for this study. I suggest that the selection and tracking of the sample be presented in the form of a flow chart too.

Reviewers' comments:

Reviewer's Responses to Questions

**Comments to the Author**

1. Is the manuscript technically sound, and do the data support the conclusions?

Reviewer #1: Yes

Reviewer #2: Yes

2. Has the statistical analysis been performed appropriately and rigorously? 

Reviewer #1: Yes

Reviewer #2: Yes

3. Have the authors made all data underlying the findings in their manuscript fully available?

Reviewer #1: Yes

Reviewer #2: Yes

4. Is the manuscript presented in an intelligible fashion and written in standard English?

Reviewer #1: Yes

Reviewer #2: Yes

5. Review Comments to the Author

Reviewer #1: Thank you for the opportunity to review this important manuscript reporting the association between GDM and future CKD. I think this is a timely submission, highlighting the potential for early intervention to reduce progression. I particularly liked the clarity of the definitions and uncertainty around diagnosis of GDM. Real world data are never perfect!

I have a few suggestions to enhance the manuscript.

1. Abstract – please add CKD Stages 4 and 5 for clarity

2. I am perplexed by the association with ‘glomerular/proteinuric CKD’ – I suspect this is most likely to be due to secondary FSGS. I assume it is not possible to have more clarity on breakdown of these conditions. What is the overall prevalence of difference glomerular diseases in the Swedish renal registry? And other conditions to allo comparison.

3. Would it be possible to report median time to CKD diagnosis and/or age of CKD diagnosis – was this different for DM / HT disease – and how does this compare to the Swedish Renal Registry? Is there a signal that women with GDM are likely to develop DKD earlier than other diabetics?

4. How many women had GDM in two pregnancies – is this a stronger signal for future risk of CKD

5. Was there an increase in prevalence of CKD with time? Or is there uncertainty about

6. Could mortality be reported? This is a very high risk group, and whilst numbers are likely to be low, early maternal death is likely to be higher in this group.

7. The authors could highlight in more detail the importance of this work – GDM affects more women of lower education, obese and migrants - there is an opportunity to intervene to reduce health inequalities

8. The authors could also discuss the implications of the finding of more severe CKD suggesting there is likely to be considerably more less severe CKD and associated cardiovascular risk. Women are less likely to have CVD diagnosed early and less likely to benefit from prevention strategies. Could pregnancy history be another opportunity for risk stratification?

Reviewer #2: Thank you for giving me the opportunity to review this manuscript.

The topic is interesting and the paper is well written.

However, the data has several issues with it. The first is that the prevalence is only 1%, meaning that there is under diagnosis. Which is also the problem because there is no screening protocol for gem.

The second is that the data goes back over 30 years which raises the chance of recall bias and under diagnosis.

The analysis also doesn’t take into account women who delivered more than once. I would suggest to only take the last five years and do the analysis again.

6. PLOS authors have the option to publish the peer review history of their article (what does this mean?). If published, this will include your full peer review and any attached files.

Reviewer #1: **Yes: **Kate Bramham

Reviewer #2: No

---

## [Author Response · Author response to Decision Letter 0]

14 Jan 2022

EDITORIAL COMMENTS

1. Thank you for this study. I suggest that the selection and tracking of the sample be presented in the form of a flow chart too.

Thank you for this suggestion. We have now included a flow chart as a supplementary figure.

REVIEWER 1 COMMENTS

1. Thank you for the opportunity to review this important manuscript reporting the association between GDM and future CKD. I think this is a timely submission, highlighting the potential for early intervention to reduce progression. I particularly liked the clarity of the definitions and uncertainty around diagnosis of GDM. Real world data are never perfect!

Many thanks for this very positive feedback on our manuscript.

2. Abstract – please add CKD Stages 4 and 5 for clarity

Apologies that this was unclear previously. We have now clarified in the Methods section of the Abstract that this study included individuals with CKD stages 3, 4 or 5. Both the Swedish Renal Register and National Patient Register may include some outcome data on CKD with eGFR 30-60 mL/min/1.73m2 i.e. CKD stage 3. It was not mandatory for the Swedish Renal Register to report patients who had CKD stage 3, and so these only made up a minority of all CKD cases, but some were present within the dataset. This has also been clarified in the Methods section of the main manuscript.

3. I am perplexed by the association with ‘glomerular/proteinuric CKD’ – I suspect this is most likely to be due to secondary FSGS. I assume it is not possible to have more clarity on breakdown of these conditions. What is the overall prevalence of difference glomerular diseases in the Swedish renal registry? And other conditions to allow comparison.

Thank you for this comment. We are also uncertain why this association might persist in the data, and secondary FSGS is a strong possibility. The categories of CKD subtypes were relatively broad, and the process for their selection is outlined in more detail in our previous referenced publication (reference 25). In brief, we selected these categories a priori based on guidance from the National Kidney Foundation, prior research, and clinical advice from Consultant Nephrologists. CKD subtype/aetiology was always based on the initial CKD diagnosis, when each woman first appeared in either the Swedish Renal Register or National Patient Register. The specific ICD codes to which these corresponded are provided in Supplemental Table S1 in the appendix. For glomerular/proteinuric CKD, these mainly corresponded to nephrotic syndrome, nephritic syndrome, and chronic glomerulonephritis. This has been clarified in the Methods section (Outcome variables).

The prevalence of different glomerular diseases in the Swedish Renal Registry can be found in the annual reports of the Swedish Renal Registry at this website: https://www.medscinet.net/snr/arsrapporter.aspx. For example, the 2021 report suggests a prevalence of 25% glomerulonephritis, 17% diabetic nephropathy, 11% adult polycystic kidney disease, 22% other forms of CKD. However, these prevalences do not correspond directly with the data presented in our study, since we excluded women with congenital and hereditary forms of renal disease at baseline (e.g. polycystic kidney disease) and the burden of some CKD subtypes may have changed over time. Moreover, we only followed women up for a median 12 years after their index delivery, whereas the Swedish Renal Registry includes data on all women (and men) regardless of their age or parity. We also supplemented our outcome data with women who had identifiable diagnoses of CKD in the National Patient Register before the Swedish Renal Register began to record such diagnoses (e.g. CKD data was only available in the Swedish Renal Register from 2007 onwards, but inpatient and outpatient diagnosis data was available from the National Patient Register before then.

4. Would it be possible to report median time to CKD diagnosis and/or age of CKD diagnosis – was this different for DM / HT disease – and how does this compare to the Swedish Renal Registry? Is there a signal that women with GDM are likely to develop DKD earlier than other diabetics?

Thank you for this suggestion. It is possible to report median time to CKD diagnosis, and we have now done so in the Results section. We have also provided more detailed results on this in the Supplement. The median time (IQR) to CKD diagnosis among those exposed to GDM was 5.9 (2.6-12.0) years and the median time to CKD diagnosis among those who were never exposed to GDM was 6.7 (2.8-11.4) years. This difference was statistically significant (log-rank p <0.001). 

The median time to diagnosis of diabetic CKD was 6.7 years (2.7-12.2). Women who were exposed to GDM were more likely to be diagnosed with diabetic CKD earlier (median 5.7 years, IQR 3.0-9.6) than women who were never exposed to GDM (median 7.0 years, IQR 2.7-12.6). However, this difference was not statistically significant (logrank p=0.66). The median time to diagnosis of other CKD subtypes varied, but it was longest for women who experienced hypertensive CKD (median 9.8 years, IQR 4.9-15.9). It was comparatively shorter for tubulo-interstitial CKD, 6.1 years (2.8-11.2); glomerular/proteinuric CKD, 5.0 years (2.2-10.3); other/unspecified CKD, 6.7 years (2.8-13.3). There were no statistically significant differences in the median times to diagnosis of these CKD subtypes by GDM exposure, but this may have been partly due to the relatively small numbers of affected women when CKD was categorised in to subtypes. Direct comparisons were not drawn with the Swedish Renal Registry because the median time to diagnosis depends on the median follow-up time of the study sample and the exclusions applied. We only included parous women in this study and followed them up for a median 12 years (beginning from 1987 for reasons of data availability and ICD coding), and a maximum 27 years. 

5. How many women had GDM in two pregnancies – is this a stronger signal for future risk of CKD

Thank you for this interesting question. Unfortunately, we were limited in our ability to investigate the impact of recurrent exposure to GDM because of our use of time-dependent covariates. We tried doing so by looking at women with exactly two previous pregnancies, as this would allow comparison across three possible categories of exposure (i.e. none, one, two affected pregnancies). This allowed us to capture the impact of recurrent exposure to GDM without introducing an unwieldy number of permutations with time-dependent covariates. However, this analytical approach led to a considerable reduction in statistical power. Of the 784,864 women who had exactly two pregnancies, 2,838 developed CKD during follow-up. Of those, only 52 (1.8%) had been diagnosed with GDM in one pregnancy, and 10 women (0.4%) had been diagnosed with GDM in both pregnancies. We checked for any potential signal of increased future risk of CKD, and there was no evidence of this. Women who had one previous diagnosis of GDM had 79% increased risk of developing CKD during follow-up (HR 1.79, 95% CI 1.36-2.35) whereas those who had two previous diagnoses of GDM had 77% increased risk of developing CKD during follow-up (HR 1.77, 95% CI 0.95-3.30). However, this analysis may have been limited by the potential under-diagnosis of GDM in our sample. 

6. Was there an increase in prevalence of CKD with time? Or is there uncertainty about this.

Thank you for this question. There is uncertainty around how the prevalence of CKD may have changed over time. However, it is likely that more cases of CKD were diagnosed in the latter years of follow-up, and that the prevalence of CKD did increase over time. The reasons for this are multifactorial, including the following: (i) improved case ascertainment due to inclusion of data from outpatient diagnoses after 2001, and inclusion of CKD data from the Swedish Renal Register from 2007 onwards (ii) advancing age of women who were followed up for longer periods in the dataset, and thus may have been at increased risk of CKD in latter years of the study (iii) evidence from the current Swedish dataset that the prevalence of pre-pregnancy obesity was increasing over time, from 3.9% prevalence in 1987-91 up to 12.1% prevalence in 2007-12 (as shown in Supplementary Table S2) and consequently that the risk of CKD may have been increasing among parous women (iv) increasing global prevalence of CKD in recent decades, with Global Burden of Disease data suggesting a 29% increase in CKD prevalence worldwide from 1990 to 2017 due to increasing life expectancy, rising prevalence of obesity, hypertension and diabetes.

7. Could mortality be reported? This is a very high risk group, and whilst numbers are likely to be low, early maternal death is likely to be higher in this group.

Thank you for this suggestion. Of the 1,121,633 unique women included in the study cohort, 12,232 (1.1%) died during follow-up. The mortality rate did not significantly differ by GDM diagnosis (GDM-diagnosed, n=154 deaths; 1.26%, vs. no GDM diagnosis, n=12,078 deaths; 1.08%) (p=0.804). By contrast, the mortality rate was significantly higher among those women who developed CKD during follow-up (n=349 deaths; 5.9%) compared with those who did not develop CKD (n=11,883 deaths; 1.1%) (p<0.001). We have now reported the mortality rate, and provided some detail on the above analysis, in the Results section.

8. The authors could highlight in more detail the importance of this work – GDM affects more women of lower education, obese and migrants - there is an opportunity to intervene to reduce health inequalities

Thank you for this suggestion, and we agree with the reviewer. We have now added detail to the end of the Discussion section to highlight the importance of this work from a public health perspective, including the opportunity it presents for informing preventive interventions and reducing health inequalities.

9. The authors could also discuss the implications of the finding of more severe CKD suggesting there is likely to be considerably more less severe CKD and associated cardiovascular risk. Women are less likely to have CVD diagnosed early and less likely to benefit from prevention strategies. Could pregnancy history be another opportunity for risk stratification?

Thank you for this suggestion and we agree with the reviewer that pregnancy history may present a further opportunity for risk stratification. We also agree that women are less likely to be diagnosed with CVD (and CKD) at an early stage of their disease and that obstetric factors may play an important role in optimising and directing preventive strategies. We have discussed this in detail in a previous editorial in another journal (Acta Obstetricia et Gynecologica, July 2020), but we have now elaborated on these points at the end of the Discussion too. 

REVIEWER 2 COMMENTS

1. Thank you for giving me the opportunity to review this manuscript.

The topic is interesting and the paper is well written.

Many thanks for this positive feedback on our manuscript

2. However, the data has several issues with it. The first is that the prevalence is only 1%, meaning that there is under diagnosis. Which is also the problem because there is no screening protocol for gem.

We agree with this comment. Unfortunately, GDM is very likely to be under-diagnosed in this study. We have acknowledged this in the fourth paragraph of our Discussion, and we acknowledge that there may be multiple reasons for this, including: lack of universal screening for GDM during the study period, insufficient uptake of screening when offered, use of relatively strict diagnostic thresholds for GDM, inclusion of a largely Caucasian sample who are likely to be relatively healthy and lower risk for GDM at baseline (given the low prevalence of maternal obesity observed in our sample). Notwithstanding these limitations, the Medical Birth Register (the main source of data on GDM) was validated in 2002, and the quality of variables was deemed to be high. Thus, we believe that the validity of GDM diagnoses is high where available. This point is referenced in the Methods section.

3. The second is that the data goes back over 30 years which raises the chance of recall bias and under diagnosis.

Thank you for this comment. While we agree with the reviewer’s concerns about the possibility of under-diagnosis of GDM (per Response 2 above), recall bias is unlikely to be a substantial issue in this study. All diagnostic data were based on hospital or registry records and this was a retrospective cohort study by design. Since 1982, all data in the Medical Birth Register in Sweden have been retrieved directly from medical records (i.e. antenatal records, delivery records, and infant examination records respectively) to prevent any possible discrepancies during data transfer and to avoid recall bias. Since our study is based entirely on data collected from 1987 onwards, information on antenatal/perinatal exposures is based on the Medical Birth Register during a time when the quality of data is believed to be high. Further information on this is available here: https://www.socialstyrelsen.se/globalassets/sharepoint-dokument/artikelkatalog/ovrigt/2003-112-3_20031123.pdf

4. The analysis also doesn’t take into account women who delivered more than once. I would suggest to only take the last five years and do the analysis again.

Thank you for this comment. While we did not specifically analyse the association between GDM and CKD according to parity, we did adjust for parity as a potential confounder in all of our multivariable models. We also undertook a sensitivity analysis which considered recurrent exposure to GDM in women who had exactly two pregnancies, as outlined above (page 2, Response 5 to Reviewer 1). Regarding the suggestion to repeat the analysis by the last five years (i.e. 2008-2012 inclusive, with follow-up to end of 2013), we have now done so in a separate sensitivity analysis. By limiting the dataset to this time period, the number of participating women is reduced to 237,200. However, while this sample remains large, it results in a substantial reduction in the number of women who were diagnosed with GDM (n=2,972) and the number of women who were diagnosed with CKD during the remaining follow-up time (n=486). Of those who were diagnosed with GDM, only 8 went on to have a CKD diagnosis within the 5-year follow-up period after applying all other exclusion criteria (i.e. women with pre-existing comorbidities, those with congenital/hereditary forms of CKD etc). This low number may be partially explained by the fact that most of these women would have had less than 5 full years of follow-up. For example, women diagnosed with GDM in 2011 would have only had a maximum of 3 years follow-up (to end 2013). It was not possible to analyse the risk of CKD subtypes (n<5 for all), and it was not possible to analyse the risk of ESKD (n<5). 

The results of the sensitivity analysis are summarised below. The observed associations were attenuated compared with the main results for the full 27 year follow-up period. However, these effect estimates had wide 95% confidence intervals since they were based on smaller numbers of observations. It is uncertain whether these results represent true effects, or whether they may have been impacted by reduced statistical power. 

Sensitivity analysis 1 – Hazard ratios for maternal chronic kidney disease by history of GDM among women whose first birth occurred between 2008 and 2012 in Sweden.

 Chronic kidney disease (N=486)

 n Age-adjusted Fully adjusted

Ever had GDM HR (95% CI) HR (95% CI)

None 478 1.0 1.0

GDM 8 1.32 (0.60-2·55) 0.81 (0.40-1.64)

Sensitivity analysis 2 – Hazard ratios for maternal chronic kidney disease by history of GDM and/or type 2 diabetes among women whose first birth occurred between 2008 and 2012 in Sweden.

 Chronic kidney disease (N=486)

 n Age-adjusted Fully adjusted

Ever had GDM or T2DM HR (95% CI) HR (95% CI)

None 463 1.0 1.0

GDM only <5 0.86 (0.25-1.86) 0.65 (0.17-1.60)

T2DM only 15 7.88 (4.71-13.17) 6.39 (0.80-10.73)

GDM + T2DM <5 13.44 (5.02-36.03) 7.88 (2.91-21.33)

---

## [Decision Letter · Decision Letter 1]

22 Feb 2022

Does gestational diabetes increase the risk of maternal kidney disease? A Swedish national cohort study

PONE-D-21-31275R1

Dear Dr. Barrett,

We’re pleased to inform you that your manuscript has been judged scientifically suitable for publication and will be formally accepted for publication once it meets all outstanding technical requirements.

Kind regards,

Forough Mortazavi

Academic Editor

PLOS ONE

Additional Editor Comments (optional):

Reviewers' comments:

Reviewer's Responses to Questions

**Comments to the Author**

1. If the authors have adequately addressed your comments raised in a previous round of review and you feel that this manuscript is now acceptable for publication, you may indicate that here to bypass the “Comments to the Author” section, enter your conflict of interest statement in the “Confidential to Editor” section, and submit your "Accept" recommendation.

Reviewer #1: All comments have been addressed

Reviewer #2: All comments have been addressed

2. Is the manuscript technically sound, and do the data support the conclusions?

Reviewer #1: Yes

Reviewer #2: Yes

3. Has the statistical analysis been performed appropriately and rigorously? 

Reviewer #1: Yes

Reviewer #2: Yes

4. Have the authors made all data underlying the findings in their manuscript fully available?

Reviewer #1: Yes

Reviewer #2: Yes

5. Is the manuscript presented in an intelligible fashion and written in standard English?

Reviewer #1: Yes

Reviewer #2: Yes

6. Review Comments to the Author

Reviewer #1: Excellent and really important paper - congratulations!

Reviewer #2: The authors answered all my previous concerns

I believe that the paper should be accepted as it is interesting and we’ll planned out

7. PLOS authors have the option to publish the peer review history of their article (what does this mean?). If published, this will include your full peer review and any attached files.

Reviewer #1: **Yes: **Kate Bramham

Reviewer #2: **Yes: **Yael Baumfeld

---

## [Editor Report · Acceptance letter]

1 Mar 2022

PONE-D-21-31275R1 

Does gestational diabetes increase the risk of maternal kidney disease? A Swedish national cohort study 

Dear Dr. Barrett:

I'm pleased to inform you that your manuscript has been deemed suitable for publication in PLOS ONE. Congratulations! Your manuscript is now with our production department. 

Kind regards, 

on behalf of

Dr. Forough Mortazavi 

Academic Editor

PLOS ONE